# Autologous Red Blood Cell Delivery of Betamethasone Phosphate Sodium for Long Anti-Inflammation

**DOI:** 10.3390/pharmaceutics10040286

**Published:** 2018-12-18

**Authors:** Xiumei Zhang, Mingfeng Qiu, Pengcheng Guo, Yumei Lian, Enge Xu, Jing Su

**Affiliations:** School of Pharmacy, Shanghai Jiao Tong University, Shanghai 200240, China; xiumeizhang@sjtu.edu.cn (X.Z.); mfqiu@sjtu.edu.cn (M.Q.); gpcedu@sjtu.edu.cn (P.G.); lym-0517@sjtu.edu.cn (Y.L.); engles@sjtu.edu.cn (E.X.)

**Keywords:** red blood cells, drug delivery system, betamethasone phosphate, long anti-inflammation

## Abstract

Although glucocorticoids are highly effective in treating various types of inflammation such as skin disease, rheumatic disease, and allergic disease, their application have been seriously limited for their high incidence of side effects, particularly in long term treatment. To improve efficacy and reduce side effects, we encapsulated betamethasone phosphate (BSP) into biocompatible red blood cells (RBCs) and explored its long acting-effect. BSP was loaded into rat autologous erythrocytes by hypotonic preswelling method, and the loading amount was about 2.5 mg/mL cells. In vitro, BSP loaded RBCs (BSP-RBCs) presented similar morphology, osmotic fragility to native RBCs (NRBCs). After the loading process, the loaded cells can maintain around 70% of Na^+^/K^+^-ATPase activity of natural cells. In vivo, a series of tests including survival, pharmacokinetics, and anti-inflammatory effect were carried out to examine the long-acting effect of BSP-RBCs. The results shown that the loaded cells could circulate in plasma for over nine days, the release of BSP can last for over seven days and the anti-inflammatory effect can still be observed on day 5 after injection. Totally, BSP-loaded autologous erythrocytes seem to be a promising sustained releasing delivery system with long anti-inflammatory effect.

## 1. Introduction

Glucocorticoids are one kind of the most effective drugs for treating various inflammation such as skin disease, rheumatic disease, and allergic disease. However, their system applications should be used only when necessary and in as low a dose as possible for their high incidence of adverse effects, particularly in long term treatment. After being administered intravenously, glucocorticoids can be quickly cleared from body, thus to maintain effective drug concentration, several injections are recommended in one day. Many studies have reported that frequent and high dosing of glucocorticoids will cause hormone-dependence and serious side effects, such as immunological suppression and diabetes [1,2,3,4]. To avoid rapid fluctuations in blood, it is necessary to develop a novel delivery system for the sustained releasing of glucocorticoids.

Betamethasone is a kind of classic glucocorticoids which plays an important role in controlling many inflammations. To prolong the release of betamethasone phosphate (BSP), the administration of drug encapsulated in biodegradable polymeric particles has been investigated. Ishihara et al. [5] firstly developed betamethasone disodium 21-phosphate (BP) encapsulated in nanoparticles of poly(d,l-lactic acid (PLA)) homopolymers, which had strong anti-inflammatory activity. Other nanoparticles for BSP loading also include poly(d,l-lactic/glycolic acid) (PLGA) nanoparticle, and polyethylene glycol (PEG) modified PLGA-PLA nanoparticle (PEG-PLGA-PLA), but the rapid clearance from circulation by mononuclear phagocyte system was also reported. Although the surface modification with PEG can reduce antibody opsonization and prevent interactions with the mononuclear phagocyte system, allergic reactions to these preparations still occurred [6,7]. Thus, a more biocompatible vector for BSP loading is strongly needed.

Red blood cells (RBCs) are the most abundant type of blood cells and they can circulate in humans for about three months and in mice for about 40 days [8,9,10,11,12,13,14]. And in comparison with any other carriers, they hold the advantage of higher biocompatibility and longer life-span in circulation, especially when autologous erythrocytes are used [15,16,17,18,19,20,21]. He et al. [22] have reported to load asparaginase into erythrocytes for treatment of acute lymphoblastic leukemia and accomplish its long acting effect. It is clear that if the physical and biological properties of erythrocytes can be preserved, the encapsulated drug can own the similar circulating time to normal erythrocytes. In this study, BSP loaded RBCs (BSP-RBCs) were prepared by using hypotonic preswelling method (Figure 1), and to examine the long acting effects of loaded cells, survival of the preparation in circulation, drug release in plasma and anti-inflammatory effect after injection were investigated.

## 2. Materials and Methods

### 2.1. Materials

Betamethsone phosphate and prednisolone were purchased from Innochem Company (Beijing, China). Sodium chloride, sodium pyruvate, glucose, glutaraldehyde, isoamyl acetate, and xylene were purchased from Shanghai Titan Technology Co., Ltd (Shanghai, China). HPLC grade methanol was purchased from Merck & Co., Inc. (Darmstadt, German). NHC-LC-Biotin reagent was purchased from APExBIO Technology LLC (Houston, TX, USA). FITC streptavidin was obtained from Nanjing Xinfan Biological Technology Co., Ltd (Nanjing, China). Purified water was obtained from a MilliQ System (Millipore, Paris, France). All other chemicals used were of analytical grade. BCA Protein Quantification Kit was purchased from BOSTER Biological Technology (Wuhan, China). Na^+^/K^+^-ATPase Kit, Annexin V-FITC kit were both purchased from NanJing JianCheng Bioengineering Institute (Nanjing, China).

### 2.2. Animals

SD (Sprague Dawley) rats (weighing about 200 g), KM (Kunming) mice (eight-week-old females weighing 20 g) purchased from Shanghai Jiesijie Experimental Animal Co., Ltd (Shanghai, China) were used in this project. All procedures performed in studies involving animals were in accordance with the Guidelines for Care and Use of Laboratory Animals of Shanghai Jiao Tong University and approved by Institutional Animal Care and Use Committee in 19 March 2018 (Number: A2018021).

### 2.3. Preparation of BSP-RBCs

Male rats, weighing 200 g, were used throughout this experiment. Briefly, the autologous whole blood was collected into a heparinized syringe from the orbital plexus vein of rats and centrifuged at 4 °C, 600× *g* for 5 min, the plasma and buffy coat were discarded, then washed twice at 4 °C with 9 mg/mL sodium chloride (NaCl). A similar preswelling method described by Ge et al. [23] was used for encapsulation of BSP into erythrocytes. For this purpose, 0.1 mL washed packed cells were transferred gently into a microcentrifuge tube, and then resuspended in 10-times the amount of hypotonic drug solution (4.5 mg/mL NaCl, 4 mg/mL BSP) at room temperature for 20 min to swell the RBCs and load the drug. Then 0.1 mL hypertonic solution (45 mg/mL NaCl, 5 mg/mL sodium pyruvate, 10 mg/mL glucose) was added with gentle agitation, and the mixture was held at 37 °C for 30 min to reseal erythrocytes. The drug loaded RBCs obtained by this manner were finally washed with isotonic PBS three times to remove the unentrapped BSP and the released cell continents. The supernatants after resealing step and three washing steps were mixed together for drug loading amount detection.

### 2.4. Quantification of Drug

A reversed-phase HPLC method was used for drug assay. In brief, a C18 column (Ultimate XB-C18, 250 × 4.6 mm, 5 μm), a mixture of methanol and 0.05 mol/L phosphate buffer (1:1) were used as a stationary and mobile phase, respectively. The detection was performed by an ultraviolet detector in 254 nm. The collected supernatant (supernatant 1) during the drug loading procedure was used to determine the amount of unentrapped BSP. Before determining, five times the amount of methanol was added into supernatant 1 and vortex for 1 min to precipitate the released hemoglobin, then the supernatant 2 after centrifuging the mixture at 3000 g for 20 min was filtered and injected to the chromatograph. Drug loading amount (mg/mL cells) = (Total Amount − Unentrapped Amount)/0.1 mL cells. Total amount is the drug amount added in the hypotonic solution for 0.1 mL erythrocytes.

### 2.5. Scanning Electron Microscopy (SEM)

Samples from two types of erythrocytes (native RBCs (NBCs), BSP-RBCs) were prepared by fixing in glutaraldehyde (2.5%) for 30 min, incubating in the mixture of 0.4% potassium permanganate and 0.6% potassium dichromate for 30 min, and dehydrating with a concentration gradient of ethanol from 30 to 100%, each concentration for 5 min. Finally, replaced by isoamyl acetate for 30 min. The prepared samples were then analyzed using a scanning electron microscope after being coated with gold particles by a Sputter Coater in 18 mA for 30 s.

### 2.6. Osmotic Fragility

To evaluate the resistance of RBC membrane against the osmotic pressure changes of their surrounding media. Osmotic fragility was examined by incubating 0.1 mL cells into 1 mL stepwise decreasing sodium chloride solutions (0.9, 0.8, 0.7, 0.6, 0.55, 0.5, 0.45, 0.4, 0.3, 0.2, 0.1, 0% NaCl). After 20 min of incubation at room temperature, the suspensions with hemoglobin were collected by centrifugation at 600× *g* for 5 min, and then the absorbance at 540 nm of the supernatants were measured by ultraviolet spectrophotometer. The released hemoglobin was expressed as percentage of absorbance of each sample to that of a completely lysed sample prepared by incubating the cells of each type with 0% NaCl solution. For comparative purposes, an osmotic fragility index (OFI) was defined as the NaCl concentration producing 50% hemoglobin release, and 0.9% NaCl equals to osmotic pressure of 308 mOsmL^−1^ [24].

### 2.7. Activity of Na^+^/K^+^-ATPase

To investigate the possible damages of loading procedure on RBC bioactivity, the activities of Na^+^/K^+^-ATPase of NRBCs and BSP-RBCs were determined by Na^+^/K^+^-ATPase Kit. Each type of RBC pellets was lysed in ten times of distilled water. 20 μL RBC lysate was collected and diluted with 50 times of distilled water until the lysate was colorless or slightly pink. Then the activity of Na^+^/K^+^-ATPase was measured according to the protocol of Na^+^/K^+^-ATPase Kit and protein concentration was determined by using the BCA Protein Quantification Kit.

### 2.8. Phosphatidylserine(PS) Exposure

Phosphatidylserine exposure on outer membrane is the signal of early apoptosis in cells and these cells will be cleared quickly from circulation by endothelium reticular system [25]. To examine whether loading process has caused apoptosis to erythrocytes, the PS reversion rate was detected by flow cytometer [26]. Each type of RBC pellets (10 μL) were firstly resuspended in 490 μL isotonic PBS. Then 10 μL RBC pellets suspension was diluted by 100 μL 1× binding buffer and incubated with 5 μL FITC labeled annexin V for 10–15 min at room temperature in dark to label exposed PS. After incubation, 400 μL 1× binding buffer was added to stop the labeling reaction. The samples were finally analyzed with a flow cytometer (BD LSR Fortessa, Becton Dickinson, Franklin Lakes, NJ, USA) and with its accompanying software (CELLQUEST, Becton Dickinson, Franklin Lakes, NJ, USA). PS reversion rate is the ratio of FITC labeled erythrocytes.

### 2.9. Survival in Circulation

In order to study the in vivo circulation time of drug-loaded cells, we prepared the drug-encapsulated autologous erythrocytes, labeled them with biotin reagents and returned them to the rats [27]. Then we took the blood from the rats at different time points and studied the survival rate of erythrocytes by flow cytometry. Briefly, we obtained 0.7 mL of whole blood from the orbital plexus vein and collect the RBCs and plasma. Then the BSP-RBCs were prepared as described above. The RBCs and biotin reagents (15 μg/mL) were mixed at ratio of 1 to 100 and incubated at room temperature for 30 min. Cells were washed three times with PBS + 100 mM glycine to quench and remove excess biotin reagents and byproducts. Then the biotin-RBCs were suspended in autologous plasma to the volume of 0.7 mL and returned to rats by tail vein injection. 0.05 mL blood each time from rats at different time points (5 min, 1 h, 3 h, 5 h, 8 h, 24 h, 3 days, 5 days, 7 days, 9 days) was collected to get the washed erythrocytes. The washed erythrocytes and FITC streptavidin were mixed at a ratio of 1 to 100, and incubated the mixture at room temperature for 30 min. Cells were washed three times with PBS to remove excess fluorescent reagents. Finally, the cells were diluted for about 2000 times to measure the average fluorescence intensity of each sample by a flow cytometer. The average fluorescence intensity of 5 min points was considered as 100 percent of cell survival rate.

### 2.10. Pharmacokinetics (PK)

The PK of betamethasone in BSP-RBCs was determined by UPLC-MS. Firstly, the blood was taken from rats and the drug-loaded erythrocytes were prepared and returned back to rats. Then 0.1 mL blood was collected from each rat for the measurement of betamethasone concentrations at 5, 15, and 30 min and 1, 2, 4, 6, 8, 12, and 24 h after the start of infusion. Additional samples were obtained on days 2, 3, 4, 5, 6, and 7. Plasma concentration of betamethasone were determined using a validated UPLC-MS. In both groups, each individual rat was given 0.7 mL preparations with 350 μg BSP (*n* = 6 each). Standard PK parameters for BSP were calculated with Pksolver, a Microsoft Excel add-in application [28,29].

### 2.11. In Vivo Anti-Inflammatory Effect

The anti-inflammatory effect of BSP-RBCs was evaluated by using mice ear edema model. The mice (20 g) were injected intravenously firstly with BSP-RBCs (500 μg/mL, 0.1 mL). Then, at 30 min after drug injection, xylene was painted on the right ear of mice (30 μL/ear) to cause ear inflammation. After 30 min, six mice were selected and sacrificed in each group and their ears were punched (8 mm) and weighted. The same model and measurements were repeated on days 1, 3, 5, and 7 after drug injection. The differences between the weights of two ears represents the degree of inflammation, and the smaller difference means better anti-inflammatory effect.

### 2.12. Statistical and Data Analyses

Data expression was shown as the mean ± SD. Significant differences between NRBCs and BSP-RBCs were analyzed by the Tukey–Kramer multiple comparison test, using GraphPad Prism Software, v.6.01 (GraphPad Software Inc., San Diego, CA, USA). Results with *p* < 0.05 were considered statistically significant.

## 3. Results and Discussion

### 3.1. Quantification of Drug

The loading amount of the prepared BSP-RBCs was about 2.5 mg/mL cells, which was promising for the treatment of inflammatory [30]. 1 mL cells were collected from 2–3 mL whole blood by centrifuging at 600× *g* for 5 min.

### 3.2. Scanning Electron Microscopy

The morphology of the erythrocytes plays a crucial role in their life-span in circulation [31]. We investigated the possible morphological changes of erythrocytes upon loading process by using SEM. As illustrated in Figure 2, the BSP loaded erythrocytes resulted in the formation of cup-form similar to the normal erythrocytes. These findings show that the loading process may has no deleterious effects on erythrocyte shape.

### 3.3. Osmotic Fragility

The osmotic fragility test was used to detect structural changes in RBC membranes subjected to osmotic stress. The fragile cells maybe destroyed and eliminated quickly from circulation by macrophages [32,33]. As the hemolysis rate curve displayed in Figure 3, BSP-RBCs were easier to hemolysis under pressure of 103 to 205 mOsm/L, the ability of BSP-RBCs to resist osmotic pressure decreased. It suggested that drug loading made cells more fragile, which was in agreement with those of GI Harisa et al. who have found that osmotic fragility of loaded cells is higher than unloaded cells [34]. And this may be due to multiple changes in cell morphology during drug loading process and subsequent several washing steps. Although there was small decrease of osmotic fragility after loading process, the OFIs of BSP-RBCs (148 mOsm/L) and NRBCs (173 mOsm/L) were very close, the gap between them was much less than what have been reported by Hamidi et al. [35].

### 3.4. Activity of Na^+^/K^+^-ATPase

Erythrocyte membrane enzyme is an important membrane-bound enzyme, which plays an important role in keeping RBCs morphology, structure and function. Among them, Na^+^/K^+^-ATPase mainly involves in the transmembrane transport of Na^+^ and K^+^ to maintain proper iron concentration, and it is related to cells deformability and blood viscosity [36,37]. From Figure 4, Na^+^/K^+^-ATPase activity of BSP-RBCs was around 21.1 ± 2.8 μmolpi·gHb^−1^·h^−1^, which kept about 70% of NRBCs (31.6 ± 3.2 μmolpi·gHb^−1^·h^−1^). The decrease may be caused by multiple centrifugation, PBS washing, heparin anticoagulation, blood–vapor interface damage, and cells pre-swelling during the drug-loading process. The decreasing ATPase activity may affect its circulation time in vivo.

### 3.5. Phosphatidylserineexposure

After the encapsulation procedure, the presence of PS in the outer lipid layer of erythrocytes membrane was studied by annexin V to assess the membrane damage. From Figure 5A, we could see that BSP-RBCs had the similar distribution to NRBCs on FSC-SSC scatter plot, which indicated the similar morphology and confirms the results obtained from SEM. Flow cytometric analysis of BSP-RBCs carried out just after the loading procedure showed PS externalization values of about 26.0 ± 3.4% (Figure 5B), which displayed significant difference compared to NRBCs (0.4 ± 0.1). Once PS is exposed, the RBCs were selectively recognized by PS receptors present on the phagocytic cell membrane and actively ingested. Staedtke et al. reported that PS ratios >35% resulted in >90% uptake efficacy, and a correlation between uptake and PS exposure could be observed with ratios <35% [38]. Thus the 26.0 ± 3.4% exposure rate of BSP-RBCs may affect its circulation time which needs to be investigated in vivo.

### 3.6. Survival of BSP-RBC

The half-life shows the elimination rate of RBCs [39]. To study the life-span of RBCs, autologous NRBCs and BSP-RBCs were firstly labeled with biotin and then injected back to the rats. Five minutes (time zero) after injection, took the blood and incubated the RBCs with FITC, the fluorescence was measured and seen as 100% fluorescence. As shown in Figure 6A, the survival curve of NRBCs and BSP-RBCs were both characterized by an initial rapid decrease in the first 24 h, and followed by a slow but prolonged phase of biotin-RBCs elimination from plasma. This phenomenon indicated that the more damaged cells would be removed quickly from circulation, whereas the less damaged cells could be seen circulating in blood for a longer time period. Furthermore, we could also see that the detected circulation time of NRBCs was much shorter than its theoretical life span (about 40 days), the significant difference may be caused by RBCs collection and reintroduction or biotinylation.

The stretched time scale for the first 24 h in Figure 6B permitted to observe more clearly how the BSP-RBCs were removed from circulation faster than NRBCs. These findings agreed well with the vitro results shown in Figure 4 and Figure 5 that more damaged cells existing in BSP-RBC preparations. As can be also seen in Figure 6B, only 26.3 ± 4.27% of loaded cells were cleared from circulation after 24 h, which showed better survival ability than what was reported by Carmen et al. [39]. In the later days, BSP-RBCs showed similar clearance rate to NRBCs, about 15% of injected cells one day. In addition, the survival rate of BSP-RBCs was still about 27.8 ± 1.5% on day 9 which suggested the long- circulation potential of RBC.

### 3.7. Pharmacokinetics

The PK properties of BSP released from the RBCs were determined in rats who received free BSP solution and autologous BSP-RBCs preparation (*n* = 6). The mean encapsulated BSP dose was 0.35 mg in two groups. A log means plasma concentration vs time profile for both groups was shown in Figure 7 and a detailed summary of the PK parameters for betamethasone for two preparations was shown in Table 1. It was noted that the release of free BSP was rapid after the single vein injection and it almost could not be detected after 24 h, while the release of BSP in BSP-RBCs group could still be detected up to seven days. The elimination half-life of BSP in BSP-RBCs preparations was about 3.31 days, which indicated a sustained release effect, when compared with free BSP preparation, 0.15 day. The decline of betamethasone in erythrocytes was approximately similar to the survival patterns of cells.

### 3.8. In Vivo Determination of Anti-Inflammatory Effect

The measurement of anti-inflammatory effect was carried out on 30 min and days 1, 3, 5, and 7 after the administration of 0.1 mL autologous BSP-RBCs (50 μg BSP). At different time points, 30 μL Xylene was dropped onto the surface of the mice right ear for 30 min to cause ear swelling. As shown in Figure 8, at 30 min and on days 1, 3, and 5, the swelling rate of BSP-RBCs group was significantly lower than that of saline group, indicating that there was enough plasma BSP for powerful anti-inflammatory effect. However, the swelling rate of BSP-RBCs group was almost similar to the saline group on day 7 while the BSP group only showed significant difference at 30 min. Thus, betamethasone loaded in erythrocytes appeared to show sustained-release effect and the anti-inflammatory action could last for more than five days. This prolonged effect agreed reasonably well with the survival and pharmacokinetics shown in Figure 6 and Figure 7. Despite the presence of drugs in plasma on day 7, it could not meet the effective dose.

## 4. Conclusions

In this study, betamethasone phosphate was successfully loaded into autologous erythrocytes by hypotonic preswelling method, the loading amount was about 2.5 mg/mL cells. The vitro characterizations proved the similarity of BSP-RBCs to NRBCs in morphology and osmotic fragility. Although the decreased activity of ATPase and more exposure of PS proved the damages to erythrocytes caused by loading procedure, the experiments in vivo indicated the long acting ability of BSP-RBCs. In summary, the autologous erythrocytes are promising drug carriers for sustained releasing and thus improving the therapeutic outcome and decreasing the adverse effect of glucocorticoids, such as betamethasone. In addition, editing patients’ own cells to kill tumor cells like Chimeric Antigen Receptor T-Cell Immunotherapy (CAR-T) can avoid allogeneic rejection, the use of autologous red blood cells for drug delivery can also improve medication safety and provide some advice for personalized medicine.

## Figures and Tables

**Figure 1 pharmaceutics-10-00286-f001:**
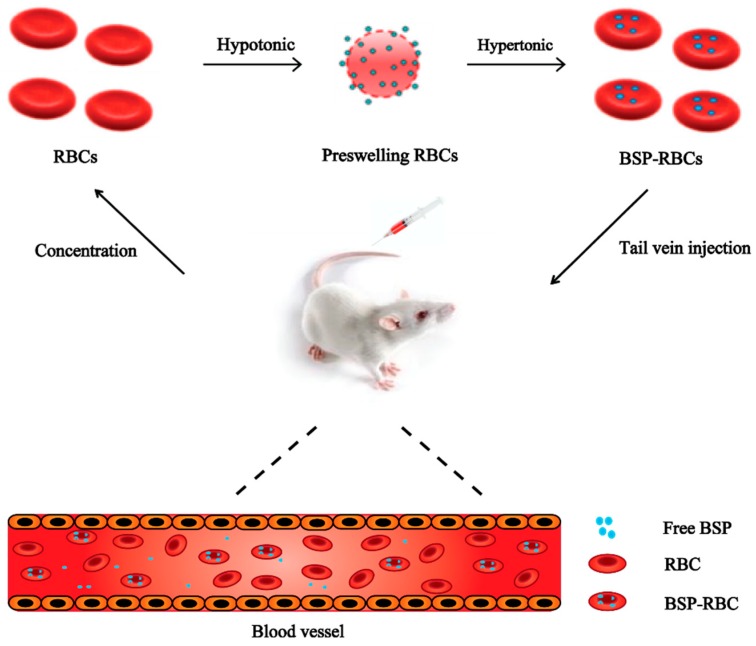
Drug loading scheme.

**Figure 2 pharmaceutics-10-00286-f002:**
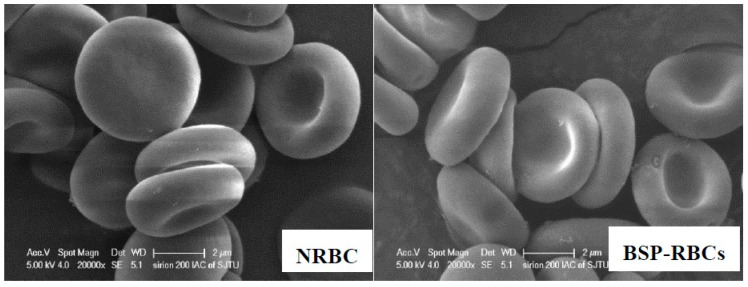
Scanning electron microscopy of NRBCs and BSP-RBCs. NRBCs have a normal biconcave shape, BSP-RBCs have the similar biconcave shape. Magnification is 20,000.

**Figure 3 pharmaceutics-10-00286-f003:**
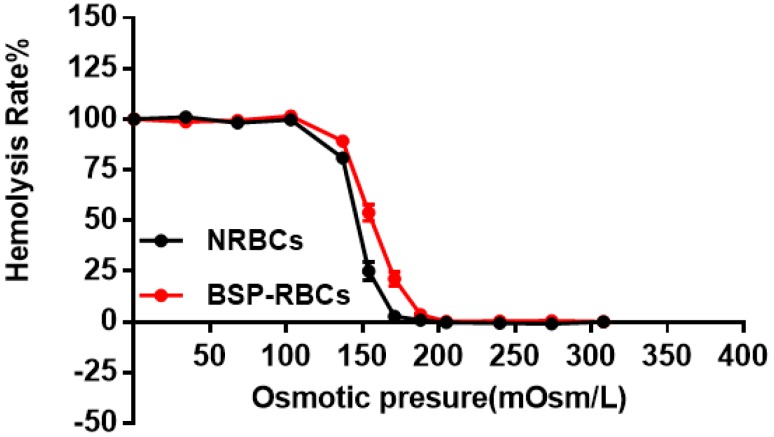
Osmotic fragility curves of NRBCs and BSP-RBCs. Data are represented as mean ± SD, three samples were used in each group.

**Figure 4 pharmaceutics-10-00286-f004:**
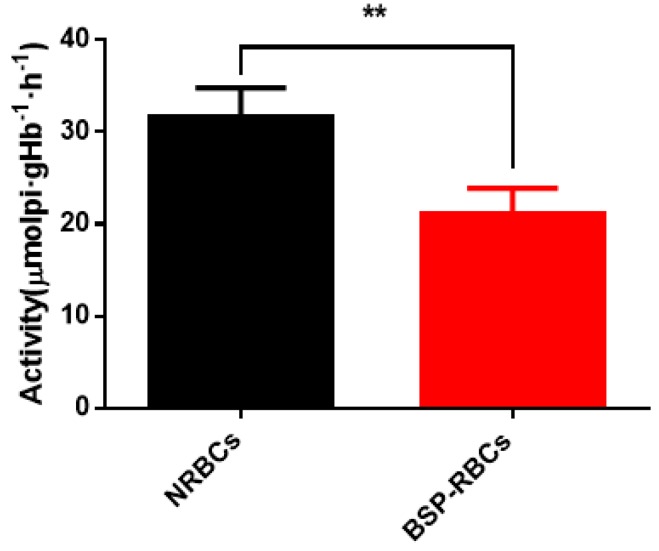
Na^+^/K^+^-ATPase activity on membrane of NRBCs and BSP-RBCs. Data are represented as mean ± SD, three samples were used in each group. ** correspond to *p* < 0.01.

**Figure 5 pharmaceutics-10-00286-f005:**
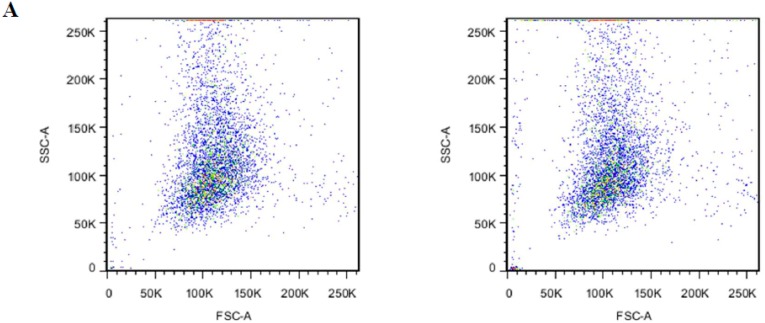
PS surface exposure of NRBCs and BSP-RBCs. (**A**) FSC-SSC scatter plot; (**B**) PS reversion rate. Data are represented as mean ± SD, three samples were used in each group. ** correspond to *p* < 0.01.

**Figure 6 pharmaceutics-10-00286-f006:**
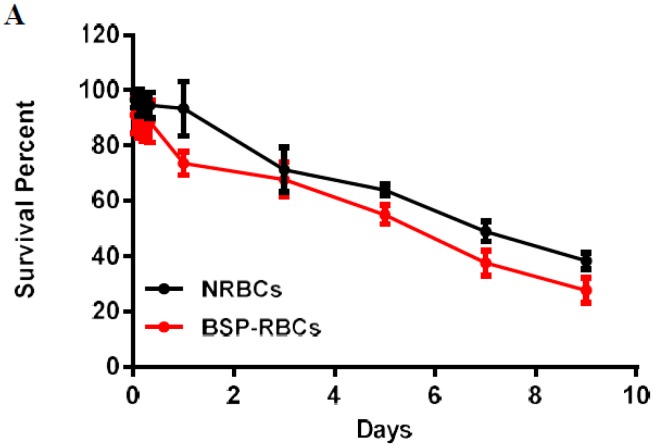
In vivo survival of NRBC and BSP-RBC. (**A**) Survival in 9 days; (**B**) Survival in the first 24 h. Data are represented as mean ± SD, three samples were used in each group.

**Figure 7 pharmaceutics-10-00286-f007:**
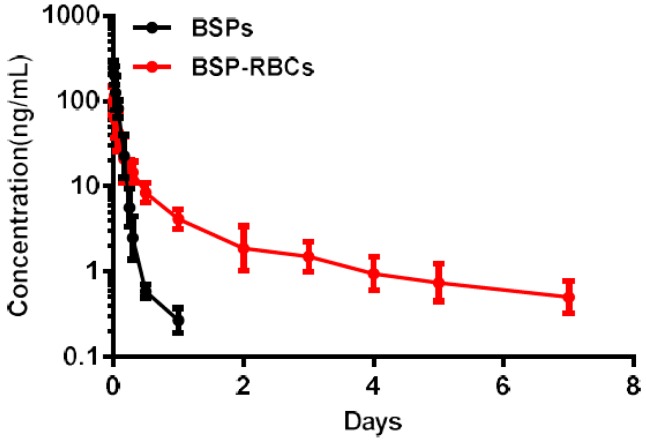
Pharmacokinetics of NRBCs and BSP-RBCs. Data are represented as mean ± SD, six samples were used in each group.

**Figure 8 pharmaceutics-10-00286-f008:**
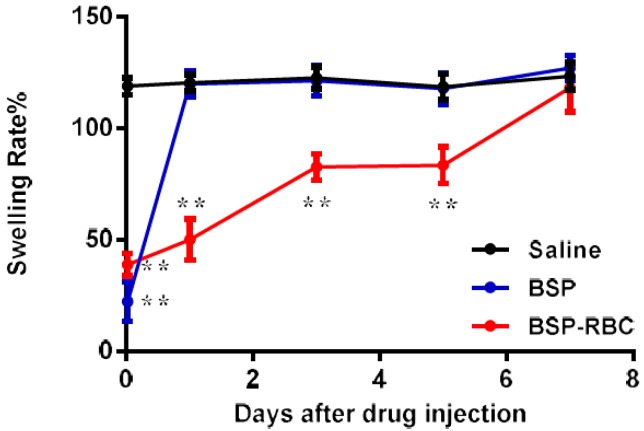
In vivo determination of anti-inflammatory effect. Data are represented as mean ± SD, six samples were used in each group. ** correspond to *p* < 0.01.

**Table 1 pharmaceutics-10-00286-t001:** Summary PK parameters for betamethasone. (mean ± SD, *n* = 6).

Preparations	*T*_1/2_ (d)	*C*_max_ (ng/mL)	AUC_0–t_ (ng/mL*d)	AUC_0–∞_ (ng/mL*d)
BSP	0.15 ± 0.03	245.77 ± 23.09	18.13 ± 4.67	18.17 ± 4.67
BSP-RBC	3.31 ± 0.82	101.40 ± 17.83	23.81 ± 6.13	26.21 ± 6.75

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
