# Peer review of "Autologous Red Blood Cell Delivery of Betamethasone Phosphate Sodium for Long Anti-Inflammation"

_pharmaceutics, 2018, doi:10.3390/pharmaceutics10040286_

Reviewer 1 Report

Comments

Major comment

The authors have proposed the encapsulation of betamethasone phosphate (BSP) into biocompatible RBCs (red 15 blood cells) as a new and innovative drug delivery system. The amount of drug loading (2.5mg/mL cells) is considered suitable for BSP in vivo use. The protocols were designed using rat autologous erythrocytes. The in vivo tests included survival, pharmacokinetics and anti-inflammatory efficacy. The results have shown that loaded-cells could circulate in plasma for over 9 days, the release of BSP can last for over 7 days and the anti-inflammatory effect can still be observed on days 5 after injection. The system proposed by the authors is a promising drug delivery system to be applied as a “personalized medicine alternative”.

The manuscript is well written, They have established a clear innovative system and demonstrated the "state of art" for biodegradable polymeric nanoparticles, and how the RBCs improve the BSP in vivo release. The manuscript can be accepted after minor revisions. Please find bellow some comments.

Major comments:

(i)          The authors could improve the introduction if this is the first study using RBCs as a drug carrier.

(ii)        I think that They could addresses some examples of personalized medicine and how the proposed system affected this.

(iii)     Please improve references with specific issue

(iv)     I have doubts about the calculations for PK experiments. I think that the RBCs is much better that BSP. This larger difference identified in the Fig 7 is not observed in Table 1.

Author Response

Point 1:  The authors could improve the introduction if this is the first study using RBCs as a drug carrier.

Response 1: Thanks for your nice suggestion. This is not the first study using RBCs as a drug carrier, but this is the first study using RBCs as a carrier for betamethasone phosphate delivery.

Point 2: I think that They could addresses some examples of personalized medicine and how the proposed system affected this.

Response 2: Thanks so much for your nice suggestion. In the reversed manuscript, we have commented on this in the conclusion part.

Point 3: Please improve references with specific issue.

Response 3: Thanks so much for your nice suggestion. We have done some revision in the manuscript.

Point 4: I have doubts about the calculations for PK experiments. I think that the RBCs is much better that BSP. This larger difference identified in the Fig 7 is not observed in Table 1.

Response 4: Thanks so much for your nice suggestion. The detailed PK parameters shown in table 1 were calculated by software PKsolver version 2.0. As can be seen in table 1, the half-life of free BSP is 0.15±0.03 d, and the half-life of BSP-RBC is 3.31±0.82d which is about 22 times of that of free BSP.

Reviewer 2 Report

Summary:

This manuscript shows the loading of betamethasone phosphate (BSP), an anti-inflammatory glucocorticoid, into red blood cells (RBC) using a hypotonic pre-swelling method to prolong therapeutic efficacy and reduce negative side effects. Overall, I found this approach to be well-characterized with sufficient data to make it interesting to readers.

Major comments:

1.       Increased osmotic fragility, decreased ATPase activity, and phosphatidylserine exposure are hypothesized as the reason why RBCs only circulate for 9 days instead of 3-4 months, like normal RBCs. However, since you can see a similar rapid drop over 9 days of the NRBCs, this would suggest that the issue is either with RBC collection and reintroduction or biotinylation. Can you please comment on this in the text?

2.       In your circulation study, what percentage of theoretical biotinylated RBCs were present at 5 minutes? Is it possible that a large number of cells were cleared during the first passage (1 minute)? If theoretical is very different than the amount measured at 5 minutes, it could be that you are normalizing to a number that is too low, which would explain why you see a lower amount of clearance than expected (26.3%).

3.       It would be interesting to know more about the proposed mechanism of BSP release from RBCs. Is it hypothesized to be simply diffusion-mediated? If so, it would seem that the process appears to be quite slow.

Minor comments:

1.       Needs significant editing for grammar beginning with the title. There are also odd spacing issues and periods where they do not belong.

2.       On line 72, please define “SD” and “KM.” Also, please define all abbreviations in the text even if they are defined in the abstract, such as “NRBCs.” Similarly, PS is never defined, but presumed to be phosphatidylserine.

3.       In Figure 2, “BSP-“ is a confusing label for BSP-loaded RBCs, please use something like “BSP-RBC as used in other figures.

Author Response

Point 1: Increased osmotic fragility, decreased ATPase activity, and phosphatidylserine exposure are hypothesized as the reason why RBCs only circulate for 9 days instead of 3-4 months, like normal RBCs. However, since you can see a similar rapid drop over 9 days of the NRBCs, this would suggest that the issue is either with RBC collection and reintroduction or biotinylation. Can you please comment on this in the text?

Response 1: Thanks so much for your nice suggestion. In this study, we collected erythrocytes from rats, and the circulation time of it is about 40 days instead of 3-4 months for humans. And we have added relevant comments in the reversed manuscript.

Point 2: In your circulation study, what percentage of theoretical biotinylated RBCs were present at 5 minutes? Is it possible that a large number of cells were cleared during the first passage (1 minute)? If theoretical is very different than the amount measured at 5 minutes, it could be that you are normalizing to a number that is too low, which would explain why you see a lower amount of clearance than expected (26.3%).

Response 2: Thanks so much for your nice question. In the circulation study, we chose the samples drawn at 5 minutes to represent the 100% percentage of biotinylated RBCs. For little clearance of RBCs by endothelium reticular system during short time, when using biotin label to study the circulation time of RBCs, the first sample was often drawn within 1 to 2 hours after reintroduction to represent the initial density of biotinylated RBCs. For example, Donald M. Mock et al chose 20 minutes as the first time point. (Reference: Red Cell Volume Can Be Accurately Determined in Sheep Using a Non-radioactive Biotin Label. Pediatr Res. 2008)

Point 3: It would be interesting to know more about the proposed mechanism of BSP release from RBCs. Is it hypothesized to be simply diffusion-mediated? If so, it would seem that the process appears to be quite slow.

Response 3: Thanks so much for your nice suggestion. We think there maybe two mechanisms of BSP release from RBCs: diffusion and leakage. BSP can be slowly converted into betamethasone by phosphatase in RBC and diffuse through phospholipid bilayer, the release speed is slow. The leakage is caused by the aging and clearance of RBCs, and the release speed is quick.

For other minor comments

(1)Needs significant editing for grammar beginning with the title. There are also odd spacing issues and periods where they do not belong.

(2) On line 72, please define “SD” and “KM.” Also, please define all abbreviations in the text even if they are defined in the abstract, such as “NRBCs.” Similarly, PS is never defined, but presumed to be phosphatidylserine.

(3) In Figure 2, “BSP-“ is a confusing label for BSP-loaded RBCs, please use something like “BSP-RBC as used in other figures.

Response: Thanks so much. And we have revised the manuscript clearly according to your suggestions.